**Data Availability Statement:** All data are available at Mendeley: Jalenques I, Cyrille D, Derost P, et al (2020) "The GTS-QOL (Gilles de la Tourette

# Cross-cultural adaptation and psychometric evaluation of the French version of the Gilles de la Tourette Syndrome Quality of Life Scale (GTS-QOL)

Isabelle Jalenques[1]*, Diane Cyrille[1], Philippe Derost[2], Andreas Hartmann[3], Sophie Lauron[1], Clara Jameux[1], Urbain Tauveron-Jalenques[1], Candy Guiguet-Auclair[4], Fabien Rondepierre[1], for The Syndrome de Gilles de La Tourette Study Group[¶]

1 Centre de Compétence Gilles de la Tourette, Service de Psychiatrie de l'Adulte A et Psychologie Médicale, CHU Clermont-Ferrand, Université Clermont Auvergne, Clermont-Ferrand, France, 2 Service de Neurologie, CHU Clermont-Ferrand, Clermont-Ferrand, France, 3 Assistance Publique-Hôpitaux de Paris (AP-HP), Pitié-Salpêtrière Hospital, National Reference Center for Tourette Syndrome, Paris, France, 4 Université Clermont Auvergne, CHU Clermont-Ferrand, CNRS, SIGMA Clermont, Institut Pascal, Clermont-Ferrand, France

¶ Membership of The Syndrome de Gilles de La Tourette Study Group is provided in the Acknowledgments.
* ijalenques@chu-clermontferrand.fr

## Abstract

### Introduction

The Gilles de la Tourette Syndrome–Quality of Life Scale (GTS-QOL) is a self-rated disease-specific questionnaire to assess health-related quality of life of subjects with GTS.

Our aim was to perform the cross-cultural adaptation of the GTS-QOL into French and to assess its psychometric properties.

### Methods

The GTS-QOL was cross-culturally adapted by conducting forward and backward translations, following international guidelines. The psychometric properties of the GTS-QOL-French were assessed in 109 participants aged 16 years and above with regard to factor structure, internal consistency, reliability and convergent validity with the MOVES (Motor tic, Obsessions and compulsions, Vocal tic Evaluation Survey) and the WHOQOL-BREF (World Health Organization Quality of Life Brief).

### Results

Exploratory factor analysis of the GTS-QOL-French resulted in a 6-factor solution and did not replicate the original structure in four subscales. The results showed good acceptability (missing values per subscale ranging from 0% to 0.9%), good internal consistency (Cronbach's alpha ranging from 0.68 to 0.94) and good test–retest reliability (intraclass correlation coefficients ranging from 0.70 to 0.81). Convergent validity with the MOVES and WHOQOL-BREF scales showed high correlations.

Syndrome–Quality of Life Scale): cross cultural evaluation of the French version", Mendeley Data, V3. doi: 10.17632/k69bbgdbbm.3.

**Funding:** IJ received a grant from the University Hospital of Clermont-Ferrand (AOI 2012) and the Association Française Syndrome Gilles de la Tourette (AO AFSGT 2015). The funders had no role in study design, data collection and analysis, decision to publish, or preparation of the manuscript.

**Competing interests:** The authors have declared that no competing interests exist.

## Discussion

Our study provides evidence of the good psychometric properties of the GTS-QOL-French. The cross-cultural adaptation and validation of this specific instrument will make it possible to assess health-related quality of life in French-speaking subjects with GTS. The GTS-QOL-French could be recommended for use in future research.

## Introduction

Gilles de la Tourette syndrome (GTS) is a neuropsychiatric disorder characterized by multiple motor and one or more vocal tics that wax and wane in frequency but have persisted for more than one year since the first tic onset [1]. A recent population-based study found a prevalence of diagnosed GTS in adults of around 0.1% [2]. Some studies report that adults with GTS perceive their health-related quality of life (HRQoL) as poor [3–5]. Early studies used a variety of generic quality of life scales [6–9]. However, generic questionnaires do not collect information on important aspects of HRQoL that apply to subjects with GTS. Disease-specific questionnaires are more relevant and more clinically sensitive as they include items focused on issues or aspects of health commonly affected by the disease [10]. A self-rated disease-specific questionnaire assessing HRQoL for subjects with GTS, the Gilles de la Tourette Syndrome–Quality of Life Scale (GTS-QOL), has been developed and validated in English [11].

The original GTS-QOL comprises 27 items referring to the past 4 weeks and rated on a 5-point Likert scale, from 0 (no problem) to 4 (extreme problem), grouped into four subscales: 'Psychological' (11 items), 'Physical and activities of daily living' (7 items), 'Obsessive-compulsive' (5 items), and 'Cognitive' (4 items). Subscale scores are calculated by adding the individual scores of the items making up the subscale, and then normalized to a 0–100 range, with higher scores representing lower HRQoL. The questionnaire also includes a visual analogue scale (VAS) ranging from 0 (representing extreme dissatisfaction with life) to 100 (extreme satisfaction). The GTS-QOL can be completed in 15 minutes or less. The questionnaire can be downloaded free of charge from the original article by Cavanna et al (2008) [11]. The psychometric properties of the questionnaire were studied among 136 subjects with GTS with a mean age of 25.9 years (SD 16.6) [11]. It had good acceptability, validity (inter-subscales correlations ranging from 0.5 to 0.7), internal consistency (Cronbach's alpha $\geq$ 0.8) and test-retest reliability (intraclass correlation coefficients $\geq$ 0.8).

The GTS-QOL is recommended in the European Clinical Guidelines for Tourette Syndrome and Other Tic Disorders for the assessment of HRQoL in adults with GTS [12]. However, studies in which HRQoL of adults with GTS is assessed by a disease-specific questionnaire as primary outcome measure are scarce. To our knowledge, there has been only one study in France whose aim was to assess HRQoL of adults with GTS, but it used a generic questionnaire [8].

Thus, we decided to perform a study to cross-culturally adapt and validate a French version of the GTS-QOL assessing its psychometric properties, to better understand HRQoL in subjects with GTS aged 16 years and above living in France. Associations between sociodemographic and clinical characteristics and HRQoL assessed by the French version of the GTS-QOL were also evaluated.

## Methods

After obtaining ethical approval, the study was conducted in two phases: 1) cross-cultural adaptation of the GTS-QOL into French and 2) psychometric properties evaluation of the GTS-QOL-French.

### Phase 1—Translation and cross-cultural adaptation of the French version of the GTS-QOL

The GTS-QOL was adapted from English into French following established international guidelines [13, 14]. First, the forward-backward procedure was applied to translate the GTS-QOL from English to French (Stage 1: scale translation). Three informed bilingual translators (2 psychiatrists and 1 neurologist) experienced in GTS research and one bilingual translator naive to the outcome measure independently translated the questionnaire into French ("forward translation"). All had French as their mother tongue and are fluent in English. The resultant version was translated back to English ("backward translation") by a native English professional translator fluent in French, blinded to the original English version and with a medical background. The authors compared the translated versions with the original English version to yield the linguistic validation of the provisional questionnaire in French. They discussed item-translation, semantic, idiomatic, experiential, and conceptual equivalents. In order to check the French-speaking adult's understanding and interpretation of the translated items, the questionnaire was pre-tested on 10 adults with GTS (clinical sample). The results were discussed between experts and patients.

Secondly, this scale was adapted for adolescents (Stage 2: scale adaptation) by three clinicians with expertise in the management of adolescents with GTS. They suggested how to simplify and rephrase items that they considered to be confusing for adolescents. The questionnaire was administered to 10 adolescents. They were asked to comment on the comprehensibility of the items. Their responses and comments were reviewed. The same experts did not suggest any wording adjustments. The French version was tested on a sample of GTS subjects (4 adults and 3 adolescents) to evaluate the comprehensibility of instructions and items. No understanding difficulty was noticed. Consequently, the expert committee adopted this version as the pre-final cross-cultural adaptation (S1 File). We named this version the GTS-QOL-French.

Thirdly, the psychometric properties of the GTS-QOL-French were examined in an independent sample of 109 people with GTS (80 males; age range: 16–64 years) (Stage 3: scale evaluation).

### Phase 2—Evaluation of the psychometric properties

**Study design.** The study was approved by the French regional ethics committee "Comité d'Ethique des Centres d'Investigation Clinique de l'inter-région Rhône-Alpes-Auvergne–CE-CIC Grenoble" (n˚ IRB 0005921, 20 September 2012). All subjects who accepted to participate received clear information on the aims and procedures of the study. All gave their written informed consent. Consent for minors, aged between16 and 18 years, was also obtained from their parents.

A set of questionnaires were given to participants during a routine consultation or mailed to them with a return envelope. To assess the test-retest reliability of the GTS-QOL-French, the questionnaire was mailed a second time to a subsample of participants (simple random sampling using random number tables) 15 days after the first assessment. Respondents who declared any change in their health status or treatment modifications since the first evaluation

or who mentioned any event that could have disrupted their quality of life between test and retest were excluded from the reliability analysis.

The sample size of the study was determined according to quality criteria established by COSMIN [15] and Terwee et al. (2007) [16] that recommend a minimum number of 100 subjects to ensure satisfactory factor analysis and internal consistency evaluation and a sample size of at least 50 subjects in order to guarantee an acceptable assessment for reliability.

**Participants.** The sample was recruited from two French specialist centres [GTS Reference Centre (Paris), GTS Competence Centre (Clermont- Ferrand)] and the "*Association Française Syndrome Gilles de la Tourette*". Participants were aged 16 years and above. They met DSM-IV-TR criteria for the diagnosis of GTS and were able to complete the questionnaires without help. A neurologist with substantial expertise in the management of GTS evaluated all participants and performed the neurological examination. The GTS subjects with a postal address who consented to participate then received a set of questionnaires to self-complete.

Of the 136 questionnaires sent to GTS subjects who had agreed to participate in the study, 109 (80%) were returned. The sociodemographic and clinical characteristics of the participants are given in Table 1. They were predominantly male (73.4%) with a mean age of 27.5 years (SD 11.5). Most declared they had tics (94.4%), medical monitoring (74.3%) and treatment for GTS (67.0%, mainly antipsychotics) in the past month.

**Study variables and instruments.** The French validated versions of the GTS-QOL, WHOQOL-BREF (World Health Organization Quality of Life Brief) and MOVES (Motor tic, Obsessions and compulsions, Vocal tic Evaluation Survey) questionnaires were self-administered.

The generic WHOQOL-BREF questionnaire comprised 26 items of which two concern overall perception of HRQoL and health, while the remaining 24 items are grouped into four subscales: 'Physical health' (7 items), 'Psychological health' (6 items), 'Social relationships' (3 items), and 'Environment' (8 items) [17]. For each subscale, scores are rated between 0 and 100, with higher values indicating better HRQoL, contrary to GTS-QOL. The WHOQOL-BREF is a cross-culturally valid assessment of HRQoL, with good to excellent reliability and validity in neuropsychiatric patients [17]. Its French version [18] has already been used very satisfactorily during a previous study in French-speaking adults with GTS [8].

The MOVES is a self-report scale measuring the severity of tics and other phenomena observed in GTS [19]. It comprises 20 items grouped into five subscales: 'Motor tics', 'Vocal tics', 'Obsessions', 'Compulsions' and 'Other associated symptoms' such as copro-, pali- and echophenomena. For each subscale, a score is obtained by adding the scores of the items listed in the subscale. A total score is calculated by adding the scores of the five subscales, which range from 0 (no symptom) to 60 (the worst condition). For clinical scoring, the 'Motor tics' and 'Vocal tics' scores are added to obtain a 'Tics' subscale score. The 'Obsessions' and 'Compulsions' scores are added to form an 'Obsessive-compulsive' subscale score. The MOVES is suggested as a severity scale for tics and related sensory phenomena and recommended as a screening instrument by the Committee on Rating Scale Development of the International Parkinson's Disease and Movement Disorder Society [20]. It has been used by Cavanna et al [11] to evaluate the GTS-QOL. A previous study in French-speaking persons with GTS aged 12 years and over provided evidence of the good psychometric properties of the French version of the MOVES [21].

Sociodemographic (age, gender, way of life, education, activity, financial aid) and medical (age at diagnosis and at onset of symptoms, tics localisation, medical monitoring, treatment, comorbidities) data were also collected. All data are available at Mendeley [22].

**Table 1. Sociodemographic and clinical characteristics of the participants.**

|  | n = 109 |
|---|---|
| Age (years), *mean (SD)* | 27.5 (11.5) |
| Male, *n (%)* | 80 (73.4) |
| Way of life, *n (%)* |  |
| Alone | 22 (20.2) |
| Couple | 34 (31.2) |
| With parents or family | 46 (42.2) |
| Institution | 7 (6.4) |
| Education, *n (%)* |  |
| Lower than high school | 50 (46.7) |
| High school diploma | 17 (15.9) |
| Higher than high school | 40 (37.4) |
| Professional activity, *n (%)* |  |
| Student | 41 (37.6) |
| Active | 40 (36.7) |
| Active in a protected environment | 5 (4.6) |
| Inactive | 23 (21.1) |
| Time since diagnosis of GTS (years), *mean (SD)* | 13.0 (8.6) |
| Age at the first symptoms (years), *mean (SD)* | 7.1 (2.8) |
| Time since the first symptoms (years), *mean (SD)* | 20.4 (11.1) |
| Subjects with tics as first symptoms, *n (%)* | 91 (86.7) |
| Location of first tics [a], *n (%)* |  |
| Face | 75 (68.8) |
| Neck | 34 (31.2) |
| Trunk | 10 (9.2) |
| Shoulder | 17 (15.6) |
| Upper limbs | 35 (32.1) |
| Lower limbs | 24 (22.0) |
| Vocal tics | 57 (52.3) |
| Others | 18 (16.5) |
| Tics in the last month, *n (%)* | 102 (94.4) |
| Current motor tics, *n (%)* | 92 (84.4) |
| Current vocal tics, *n (%)* | 74 (67.9) |
| Medical monitoring [a], *n (%)* |  |
| None | 28 (25.7) |
| General practitioner | 40 (36.7) |
| Neurologist | 57 (52.3) |
| Psychiatrist | 31 (28.4) |
| Psychologist | 11 (10.1) |
| Multidisciplinary consultation [b] | 20 (18.3) |
| Other [c] | 8 (7.3) |
| Treatment for GTS [a], *n (%)* |  |
| None | 36 (33.0) |
| Antipsychotics [d] | 62 (56.9) |
| First generation antipsychotics | 8 (7.3) |
| Second generation antipsychotics | 48 (44.0) |
| First and second generation antipsychotics | 6 (5.6) |
| Antidepressants | 23 (21.1) |
| Anxiolytics | 6 (5.5) |

(*Continued*)

**Table 1.**  (Continued)

|  | n = 109 |
|---|---|
| Hypnotics | 2 (1.8) |
| Mood stabilisers | 2 (1.8) |
| Other [e] | 17 (15.6) |
| Comorbidities, *n (%)* |  |
| None | 51 (48.1) |
| One comorbidity | 37 (34.9) |
| Two or more comorbidities | 18 (17.0) |
| Somatic comorbidities, *n (%)* | 34 (32.1) |
| Psychiatric comorbidities, *n (%)* | 33 (31.1) |

[a] A participant may have different locations for first tics, different medical monitoring, and different treatments for GTS.

[b] Neurologist and psychiatrist (*n = 13*), neurologist and psychologist (*n = 3*), neurologist and psychiatrist and psychologist (*n = 4*).

[c] Physiotherapist (*n = 2*), hypnotherapist (*n = 2*), homeopathic practitioner (*n = 1*), relaxation therapist (*n = 1*), speech therapist (*n = 1*).

[d] Mainly Ariprazole (*n = 42*), Risperidone (*n = 11*), Pimozide (*n = 9*), Haloperidol (*n = 2*).

[e] Antiepileptic (*n = 5*), Tetrabenazine (*n = 4*), antiparkinsonian (*n = 3*), psychostimulant (*n = 3*), homeopathy (*n = 3*), antihypertensive (*n = 2*), Botulinum toxin (*n = 1*), Melatonin (*n = 1*).

## Statistical analysis

Statistical analyses were performed with SAS v9.4 software (SAS Institute) and conducted at a two-sided alpha = 0.05 significance level.

**Data completeness.**   The respondent acceptability of the questionnaire was assessed by looking at the frequency of missing values. Data quality was considered satisfactory if less than 5% of the item data were missing.

**Factor analysis.**   Factor analysis with the principal axis extraction method and oblique promax rotation, allowing the factors to correlate, were performed to study the multidimensionality and distribution of the items in subscales [23]. As the perception and definition of HRQoL vary from culture to culture there was no guarantee that the French version reproduced the subscales of the original questionnaire. Hence, we chose an exploratory analysis of the structure of the items [13, 24–26]. The Kaiser-Meyer-Olkin (KMO) statistic and Bartlett's test of sphericity were used to check the appropriateness of running the factor analysis. KMO values higher than 0.5 are acceptable [27]. Bartlett's test requires to yield significant result ($p < 0.05$). Eigenvalues higher than 1 (Kaiser criterion) and Cattell's scree plot [28] were used for factor retention. The solution giving the most adequate factor structure (item loadings greater than 0.32, no or few item cross loadings, i.e. no or few items with loadings at 0.32 or higher on two or more factors) was retained [23].

**Descriptive statistics and score distributions.**   Mean, standard deviation, median, range and coefficient of skewness were used. The variability of the GTS-QOL-French scores was investigated for each subscale with the floor and ceiling effects. These effects were considered to be present if more than 15% of the subjects obtained the lowest or highest possible score [29].

**Internal consistency.**   Cronbach's α coefficient was used to evaluate the internal consistency of each subscale [30]. The minimum required for the coefficient was 0.70, according to the standard used for group comparisons [31].

**Item-total correlations.** Item-total consistency was used to evaluate the extent of the linear relationship between an item and its subscale, corrected for overlap (the item which is to be correlated with the scale was omitted from the scale total) [26]. A minimum correlation coefficient of 0.40 was considered indicative of good item-total consistency [32].

**Inter-subscale correlations.** Spearman's coefficients were used to evaluate inter-subscale correlations. Correlations were considered very small for coefficients lower than 0.30, small for coefficients between 0.30 and 0.50, moderate from 0.50 to 0.70 and strong if higher than 0.70 [33].

**Reliability.** Stability over time was assessed by the test-retest method. Reliability of the subscales was estimated by intraclass correlation coefficient (ICC), based on the two-way random effect model. Coefficients higher than 0.70 were considered satisfactory [16].

**Convergent validity.** The relationships between GTS-QOL-French and (1) WHOQOL-BREF, (2) MOVES subscales and (3) GTS-QOL-French VAS were studied, by calculating Spearman ρ correlation coefficients. Negative correlations were expected between GTS-QOL-French subscales, WHOQOL-BREF subscales, and GTS-QOL-French VAS since low scores indicated good conditions for the GST-QOL-French but bad conditions for the WHOQOL-BREF and GTS-QOL-French VAS. Positive correlations were expected between GTQ-QOL-French and MOVES subscales because low scores indicated good conditions for the two questionnaires. Correlations were considered very small for coefficients lower than 0.30, small for coefficients between 0.30 and 0.50, moderate from 0.50 to 0.70 and strong if higher than 0.70 [33].

**Associations between sociodemographic and clinical characteristics and HRQoL.** The GTS-QOL-French subscale scores were compared according to age, gender, disease duration, school attendance, professional activity, financial aid, motor and/or vocal tics, medical monitoring, comorbidities, treatment. Effect sizes (ES) were calculated by Hedges' g (a variation of Cohen's d ES that corrects for bias due to small sample sizes) and its 95% confidence interval [34]. Absolute values of 0.20 or more commonly indicated a small ES, 0.50 or more, moderate ES, and 0.80 or more, large ES [35]. For all subscales of GTS-QOL-French, a high positive ES indicated an important negative impact on HRQoL.

## Results

### Participants

The severity of tics and other symptoms, and HRQoL based on responses to MOVES and WHOQOL-BREF questionnaires, respectively, are given in Table 2.

### Data completeness

The percentage of missing values per item was low, with values ranging from 0% to 1.8%. However, 25 participants (22.9%) reported having received help to fill in the questionnaire (reading or writing).

### Factor analysis

KMO measure of sampling adequacy was 0.873 and the significance value of Bartlett's test of sphericity was <0.0001 ($\chi^2$ = 1834.7), indicating that the data were suitable for factor analysis. The factor analysis (promax rotation) of the 27 items of the GTS-QOL-French identified six factors with eigenvalues higher than one and accounting for 69.9% of the total variance (Table 3). The six subscales identified differed from the original version of GTS-QOL and were named: 'Psychological' (9 items), 'Echo-coprophenomena' (4 items), 'Social' (3 items),

**Table 2. Severity of tics (MOVES questionnaire) and health-related quality of life assessed by a generic questionnaire (WHOQOL-BREF).**

|  | Mean (SD) | Range possible |
|---|---|---|
| MOVES scores [a] |  |  |
| Motor tics | 5.7 (3.0) | 0–12 |
| Vocal tics | 2.9 (2.6) | 0–12 |
| Tics | 8.7 (4.9) | 0–24 |
| Obsessions | 2.6 (2.7) | 0–12 |
| Compulsions | 3.8 (2.4) | 0–12 |
| Obsessive-compulsive | 6.4 (4.6) | 0–24 |
| Other associated symptoms | 1.5 (2.0) | 0–12 |
| Total | 16.5 (10.0) | 0–60 |
| WHOQOL-BREF scores [b] |  |  |
| Physical health | 66.6 (18.4) | 0–100 |
| Psychological health | 54.9 (19.9) | 0–100 |
| Social relationships | 60.9 (23.6) | 0–100 |
| Environment | 72.3 (17.3) | 0–100 |

[a] Worse conditions indicated by higher scores.

[b] Worse conditions indicated by lower scores.

'Cognitive' (4 items), 'Physical/ADL' (4 items), and 'Obsessive-compulsive' (3 items). All items loaded higher than 0.32 on its subscale. Five items loaded at 0.32 or higher on two factors: item 18 loaded on factors 1 and 3 (0.37 and 0.34 respectively); item 7 on factors 2 and 6 (0.76 and 0.34 respectively); item 27 on factors 1 and 3 (0.47 and 0.58 respectively); item 2 on factors 3 and 5 (0.32 and 0.51 respectively); and item 4 on factors 2 and 5 (0.44 and 0.42 respectively). After evaluation of internal consistency and item-total correlations, items were conserved in the subscale where they loaded higher, except for item 4 which was kept in factor 5 ('Physical/ADL' subscale).

## Descriptive statistics, score distribution, floor and ceiling effects

The descriptive statistics and score distributions for the GTS-QOL-French subscales are given in Table 4. No ceiling effect was found. Important floor effects were found for 'Echo-coprophenomena' and 'Social' subscales and a slight one for the 'Obsessive-compulsive' subscale. Lower scores, corresponding to better HRQoL, were observed for the 'Echo-coprophenomena' and 'Social' subscales. Coefficients of skewness were the highest for these subscales, with a positive distribution towards good health. Higher scores, corresponding to lower HRQoL, were found for the 'Psychological' subscale.

## Internal consistency

The GTS-QOL-French subscales showed good internal consistency, with Cronbach's α ranging from 0.68 to 0.94 (Table 5). Only the 'Obsessive-compulsive' subscale did not obtain the minimum required coefficient of 0.70, having nevertheless a very close coefficient of 0.68.

## Item-total correlations

All corrected item-total correlations were higher than the required 0.40. They ranged from 0.48 to 0.81, except for item 9 with a value of 0.39, very close to that of 0.40, and indicative therefore of good item-total consistency (Table 5).

**Table 3. Factor loadings from the factor analysis of the GTS-QOL-French questionnaire.**

|  | Factor 1 | Factor 2 | Factor 3 | Factor 4 | Factor 5 | Factor 6 |
|---|---|---|---|---|---|---|
| Variance explained (%) | 41.2 | 8.8 | 5.9 | 5.4 | 4.6 | 4.1 |
| **'Psychological' subscale** |  |  |  |  |  |  |
| 16. Depressed mood | **0.59** | -0.07 | 0.28 | -0.12 | 0.07 | 0.22 |
| 17. Mood switches | **0.76** | -0.12 | -0.07 | 0.13 | 0.12 | 0.03 |
| 18. Lack of self-confidence | **0.37** | -0.06 | **0.34** | 0.21 | -0.02 | 0.08 |
| 19. Anxiety | **0.75** | -0.08 | -0.04 | 0.12 | 0.03 | 0.13 |
| 20. Restlessness | **0.77** | 0.11 | -0.18 | 0.10 | 0.13 | -0.05 |
| 21. Temper dyscontrol | **0.80** | 0.11 | -0.11 | 0.15 | 0.04 | -0.05 |
| 22. Lack of control over own life | **0.67** | 0.02 | 0.14 | -0.09 | -0.11 | 0.24 |
| 23. Frustration | **0.61** | 0.13 | 0.16 | -0.10 | 0.17 | 0.08 |
| 24. Lack of social support | **0.70** | 0.14 | 0.12 | -0.10 | 0.02 | -0.06 |
| **'Echo-coprophenomena' subscale** |  |  |  |  |  |  |
| 5. Involuntary swearing | 0.07 | **0.86** | -0.03 | -0.05 | -0.03 | -0.13 |
| 6. Embarrassing gestures | 0.04 | **0.67** | 0.05 | 0.21 | 0.04 | -0.23 |
| 7. Repeating words | 0.01 | **0.76** | 0.03 | -0.01 | -0.09 | **0.34** |
| 8. Copying people | -0.11 | **0.63** | 0.05 | 0.08 | 0.10 | 0.25 |
| **'Social' subscale** |  |  |  |  |  |  |
| 25. Difficulty seeing friends | -0.09 | -0.10 | **0.72** | 0.09 | 0.12 | 0.10 |
| 26. Difficulty taking part in social activities | 0.01 | 0.12 | **0.77** | -0.10 | 0.11 | -0.14 |
| 27. Loneliness/isolation | **0.47** | 0.03 | **0.58** | -0.07 | -0.31 | 0.01 |
| **'Cognitive' subscale** |  |  |  |  |  |  |
| 11. Difficulty concentrating | 0.24 | -0.07 | 0.27 | **0.48** | 0.14 | -0.02 |
| 12. Memory problems | -0.07 | 0.16 | 0.16 | **0.51** | 0.10 | 0.16 |
| 13. Losing important things | 0.01 | 0.06 | -0.21 | **0.78** | -0.12 | 0.16 |
| 14. Difficulty finishing tasks | 0.30 | -0.03 | 0.21 | **0.58** | -0.13 | -0.05 |
| **'Physical/Activities of daily living' subscale** |  |  |  |  |  |  |
| 1. Movement dyscontrol | 0.02 | 0.06 | 0.06 | -0.13 | **0.70** | -0.01 |
| 2. Difficulty in daily life activities | -0.02 | 0.11 | **0.32** | 0.13 | **0.51** | -0.05 |
| 3. Pain or injuries | 0.25 | -0.10 | -0.07 | 0.01 | **0.58** | 0.12 |
| 4. Phonic tics | 0.09 | **0.44** | -0.10 | -0.08 | **0.42** | 0.11 |
| **'Obsessive-compulsive' subscale** |  |  |  |  |  |  |
| 9. Repeating actions | 0.06 | 0.12 | -0.09 | 0.24 | 0.00 | **0.48** |
| 10. Unpleasant thoughts | 0.28 | 0.09 | 0.14 | 0.08 | -0.01 | **0.40** |
| 15. Concerns about poor health | 0.26 | -0.15 | 0.00 | 0.02 | 0.13 | **0.51** |

Loadings equal or higher than 0.32 are presented in bold.

In addition, each item correlated better with its parent subscale (corrected for overlap) than with the other subscales, except for two items of the 'Obsessive-compulsive' subscale (Table 5). For item 9 (repeating actions), the item-total correlation with its subscale was 0.39 and the correlation with the 'Echo-coprophenomena' subscale was 0.46. For item 10 (unpleasant thoughts), the item-total correlation with its subscale was 0.56 and the correlation with the 'Psychological' subscale was 0.63.

## Inter-subscale correlations: Spearman coefficients

Correlations between GTS-QOL-French subscales were all positive and significant ($p < 0.001$), ranging from 0.33 to 0.67 (Table 6). The 'Psychological' subscale had a small correlation with

**Table 4. Descriptive statistics and score distributions of the GTS-QOL-French subscales.**

| GTS-QOL-French subscales | Missing values (%) | Mean (SD) | Range | Median | Coefficient of skewness | Floor effect (%) | Ceiling effect (%) |
|---|---|---|---|---|---|---|---|
| Psychological | 0.9 | 43.8 (28.1) | 0–100 | 44.4 | 0.2 | 0.9 | 2.8 |
| Echo-coprophenomena | 0 | 14.7 (22.6) | 0–100 | 6.3 | 2.1 | 42.2 | 1.8 |
| Social | 0.9 | 20.9 (26.0) | 0–100 | 8.3 | 1.3 | 38.0 | 1.9 |
| Cognitive | 0 | 33.7 (25.4) | 0–100 | 31.3 | 0.6 | 9.2 | 0.9 |
| Physical/ADL | 0 | 31.9 (23.3) | 0–93.8 | 25.0 | 0.6 | 9.2 | 0 |
| Obsessive-compulsive | 0 | 28.5 (24.5) | 0–100 | 25.0 | 0.9 | 17.4 | 0.9 |

Higher scores indicate lower health-related quality of life.

ADL: activities of daily living.

the 'Echo-coprophenomena' subscale (r = 0.44) and moderate correlations with the other subscales (from 0.56 to 0.67). The 'Echo-coprophenomena' subscale had small correlations with the other subscales (from 0.39 to 0.49). The 'Social' subscale had a moderate correlation with the 'Psychological' subscale (r = 0.67) and small correlations with the others (from 0.33 to

**Table 5. Internal consistency and corrected item-total correlations for the GTS-QOL-French subscales.**

| Subscales | Cronbach's α | Corrected item-total correlations | | | | | | |
|---|---|---|---|---|---|---|---|---|
| | | Item | Psychological | Echo-coprophenomena | Social | Cognitive | Physical/ADL | Obsessive-compulsive |
| Psychological | 0.94 | 16 | **0.80** | 0.37 | 0.64 | 0.52 | 0.47 | 0.62 |
| | | 17 | **0.76** | 0.29 | 0.47 | 0.50 | 0.46 | 0.49 |
| | | 18 | **0.65** | 0.28 | 0.58 | 0.59 | 0.36 | 0.53 |
| | | 19 | **0.81** | 0.31 | 0.55 | 0.53 | 0.46 | 0.58 |
| | | 20 | **0.74** | 0.41 | 0.51 | 0.55 | 0.49 | 0.53 |
| | | 21 | **0.80** | 0.41 | 0.54 | 0.58 | 0.48 | 0.56 |
| | | 22 | **0.75** | 0.38 | 0.55 | 0.51 | 0.39 | 0.58 |
| | | 23 | **0.80** | 0.45 | 0.57 | 0.54 | 0.55 | 0.52 |
| | | 24 | **0.73** | 0.36 | 0.52 | 0.48 | 0.47 | 0.48 |
| Echo-coprophenomena | 0.83 | 5 | 0.32 | **0.70** | 0.28 | 0.30 | 0.35 | 0.31 |
| | | 6 | 0.32 | **0.61** | 0.27 | 0.40 | 0.39 | 0.26 |
| | | 7 | 0.35 | **0.66** | 0.29 | 0.33 | 0.29 | 0.41 |
| | | 8 | 0.42 | **0.65** | 0.40 | 0.34 | 0.44 | 0.37 |
| Social | 0.78 | 25 | 0.47 | 0.28 | **0.64** | 0.32 | 0.29 | 0.37 |
| | | 26 | 0.43 | 0.24 | **0.60** | 0.30 | 0.33 | 0.19 |
| | | 27 | 0.62 | 0.34 | **0.63** | 0.43 | 0.18 | 0.43 |
| Cognitive | 0.82 | 11 | 0.67 | 0.33 | 0.48 | **0.71** | 0.47 | 0.55 |
| | | 12 | 0.46 | 0.47 | 0.32 | **0.62** | 0.38 | 0.51 |
| | | 13 | 0.28 | 0.14 | 0.08 | **0.56** | 0.13 | 0.31 |
| | | 14 | 0.61 | 0.28 | 0.50 | **0.67** | 0.27 | 0.42 |
| Physical/ADL | 0.75 | 1 | 0.40 | 0.23 | 0.24 | 0.24 | **0.60** | 0.25 |
| | | 2 | 0.49 | 0.32 | 0.45 | 0.52 | **0.59** | 0.41 |
| | | 3 | 0.45 | 0.26 | 0.26 | 0.27 | **0.48** | 0.38 |
| | | 4 | 0.37 | 0.44 | 0.14 | 0.28 | **0.51** | 0.31 |
| Obsessive-compulsive | 0.68 | 9 | 0.33 | 0.46 | 0.28 | 0.35 | 0.26 | **0.39** |
| | | 10 | 0.63 | 0.36 | 0.46 | 0.52 | 0.33 | **0.56** |
| | | 15 | 0.53 | 0.29 | 0.35 | 0.39 | 0.33 | **0.53** |

Correlations of items with their parent subscale (corrected for overlap) are in bold.

ADL: activities of daily living.

**Table 6. Inter-subscale correlations and test-reliability for the GTS-QOL-French subscales.**

| GTS-QOL-French subscales | Psychological | Echo-coprophenomena | Social | Cognitive | Physical/ADL | ICC (95% CI) |
|---|---|---|---|---|---|---|
| Psychological | | | | | | 0.78 (0.58–0.89) |
| Echo-coprophenomena | 0.44 | | | | | 0.81 (0.64–0.91) |
| Social | 0.67 | 0.39 | | | | 0.71 (0.47–0.86) |
| Cognitive | 0.66 | 0.40 | 0.46 | | | 0.81 (0.64–0.91) |
| Physical/ADL | 0.56 | 0.43 | 0.33 | 0.43 | | 0.76 (0.56–0.88) |
| Obsessive-compulsive | 0.66 | 0.49 | 0.46 | 0.57 | 0.44 | 0.70 (0.45–0.84) |

Inter-scale correlations are Spearman coefficients, all significantly different from 0 (p<0.001).

ICC (95% CI): intraclass correlation coefficient (95% confidence interval).

ADL: activities of daily living.

0.46). The 'Cognitive' subscale had small correlations with 'Echo-coprophenomena', 'Social', 'Physical/ADL' subscales (r = 0.40, 0.46 and 0.43 respectively), and moderate correlations with 'Psychological' (r = 0.66) and 'Obsessive-compulsive' (r = 0.57) subscales. The 'Physical/ADL' subscale had a moderate correlation with the 'Psychological' subscale (r = 0.56) and small correlations with the others (from 0.33 to 0.44).

**Reliability.** A retest was sent to 58 participants. Of the 48 (82.8%) who returned completed questionnaires, 19 reported changes in their health status and events that had disrupted their quality of life between test and retest. Finally, 29 respondents were retained for reliability analysis. All ICCs for GTS-QOL-French subscales were greater than 0.70, ranging from 0.70 to 0.81 (Table 6).

## Convergent validity

As expected, correlations between GTS-QOL-French and WHOQOL-BREF subscales were negative (Table 7). All were significant except between the GTS-QOL-French 'Physical/ADL' and WHOQOL-BREF 'Social relationships' and 'Environment' subscales, and between the GTS-QOL-French 'Obsessive-compulsive' and WHOQOL-BREF 'Environment' subscales. Higher correlations were found between the GTS-QOL-French 'Psychological' and WHOQOL-BREF 'Physical health' and 'Psychological health' subscales (r = -0.64 and -0.67, respectively). The GTS-QOL-French 'Social' subscale was moderately correlated with WHOQOL-BREF 'Physical health' (r = -0.55), 'Psychological health' (r = -0.57) and 'Social relationships' (r = -0.61) subscales. The GTS-QOL-French 'Cognitive' subscale was moderately correlated with the WHOQOL-BREF 'Psychological health' subscale (r = -0.51). The correlation between the 'Physical' subscales of GTS-QOL-French and WHOQOL-BREF was moderate (r = -0.52). Other significant correlation coefficients ranged from -0.23 to -0.48, with the lower ones being observed for the GTS-QOL-French 'Echo-coprophenomena' subscale.

All correlations were positive and significant between GTS-QOL-French and MOVES subscales. Results showed moderate to almost strong correlations between all GTS-QOL-French subscales and the total MOVES score. Strong correlations (r ≥ 0.70) were found between the GTS-QOL-French 'Psychological' subscale and the MOVES 'Obsessions' and 'Obsessive-compulsive' subscales; between the GTS-QOL-French 'Echo-coprophenomena' and MOVES 'Other associated symptoms' subscales; between the GTS-QOL-French 'Physical/ADL' subscale and MOVES 'Motor tics' and 'Tics' subscales; and between GTS-QOL-French 'Obsessive-Compulsive' and MOVES 'Obsessions' subscales.

**Table 7. Spearman's correlation coefficients between the GTS-QOL-French and the WHOQOL-BREF, the MOVES, the GTS-QOL-French VAS subscales.**

| | GTS-QOL-French subscales | | | | | |
|---|---|---|---|---|---|---|
| | Psychological | Echo-coprophenomena | Social | Cognitive | Physical/ADL | Obsessive-compulsive |
| WHOQOL-BREF subscales | | | | | | |
| Physical health | -0.64 *** | -0.34 *** | -0.55 *** | -0.48 *** | -0.52 *** | -0.42 *** |
| Psychological health | -0.67 *** | -0.35 *** | -0.57 *** | -0.51 *** | -0.37 *** | -0.47 *** |
| Social relationships | -0.47 *** | -0.29 ** | -0.61 *** | -0.38 *** | -0.15 | -0.34 *** |
| Environment | -0.43 *** | -0.23 * | -0.46 *** | -0.40 *** | -0.13 | -0.18 |
| MOVES subscales | | | | | | |
| Motor tics | 0.43 *** | 0.41 *** | 0.24 * | 0.27 ** | 0.72 *** | 0.27 ** |
| Vocal tics | 0.51 *** | 0.64 *** | 0.40 *** | 0.43 *** | 0.54 *** | 0.38 *** |
| Tics | 0.52 *** | 0.60 *** | 0.35 *** | 0.40 *** | 0.73 *** | 0.39 *** |
| Obsessions | 0.70 *** | 0.44 *** | 0.54 *** | 0.61 *** | 0.42 *** | 0.70 *** |
| Compulsions | 0.55 *** | 0.47 *** | 0.43 *** | 0.45 *** | 0.45 *** | 0.49 *** |
| Obsessive-compulsive | 0.71 *** | 0.50 *** | 0.54 *** | 0.60 *** | 0.47 *** | 0.67 *** |
| Other associated symptoms | 0.41 *** | 0.76 *** | 0.40 *** | 0.28 ** | 0.32 *** | 0.37 *** |
| Total | 0.69 *** | 0.68 *** | 0.51 *** | 0.55 *** | 0.68 *** | 0.61 *** |
| GTS-QOL-French VAS | -0.66 *** | -0.33 *** | -0.62 *** | -0.42 *** | -0.36 *** | -0.45 *** |

Correlations significantly different from zero:

* $p<0.05$,

** $p<0.01$ and

*** $p<0.001$.

ADL: activities of daily living.

Correlations between GTS-QOL-French subscales and VAS (satisfaction with life) ranged from -0.33 with the 'Echo-coprophenomena' subscale to -0.66 with the 'Psychological' subscale. The GST-QOL-French VAS was also highly correlated with the 'Social' subscale.

## Association between sociodemographic and clinical characteristics and GTS-QOL-French subscale scores

No significant correlation was observed between GTS-QOL-French subscale scores and age or disease duration. GTS-QOL-French subscale scores were not significantly different according to gender, school attendance and physical comorbidities (Fig 1).

Working adults had a significantly better HRQoL, except for the 'Echo-coprophenomena' subscale. The negative corresponding ES were moderate for the 'Obsessive-compulsive' subscale and large for the other subscales.

Participants who were receiving a disabled adult allowance had a significantly poorer quality of life, with moderate ES for 'Physical/ADL' and 'Obsessive-compulsive' subscales and large ES for the other subscales.

Vocal tics were associated with worse HRQoL. In contrast, motor tics only (that is without vocal tics) were associated with better HRQoL except for the 'Social' subscale, on which motor tics had no significant effect. The corresponding ES were small for the 'Social' subscale, moderate for the 'Echo-coprophenomena', 'Cognitive', 'Physical/ADL' and 'Obsessive-compulsive' subscales, and large for the 'Psychological' subscale.

Participants with no medical monitoring had a significantly better HRQoL, with small ES for 'Echo-coprophenomena', 'Social', and 'Cognitive' subscales and a moderate ES for the 'Physical/ADL' subscale. In contrast, multidisciplinary medical monitoring was associated

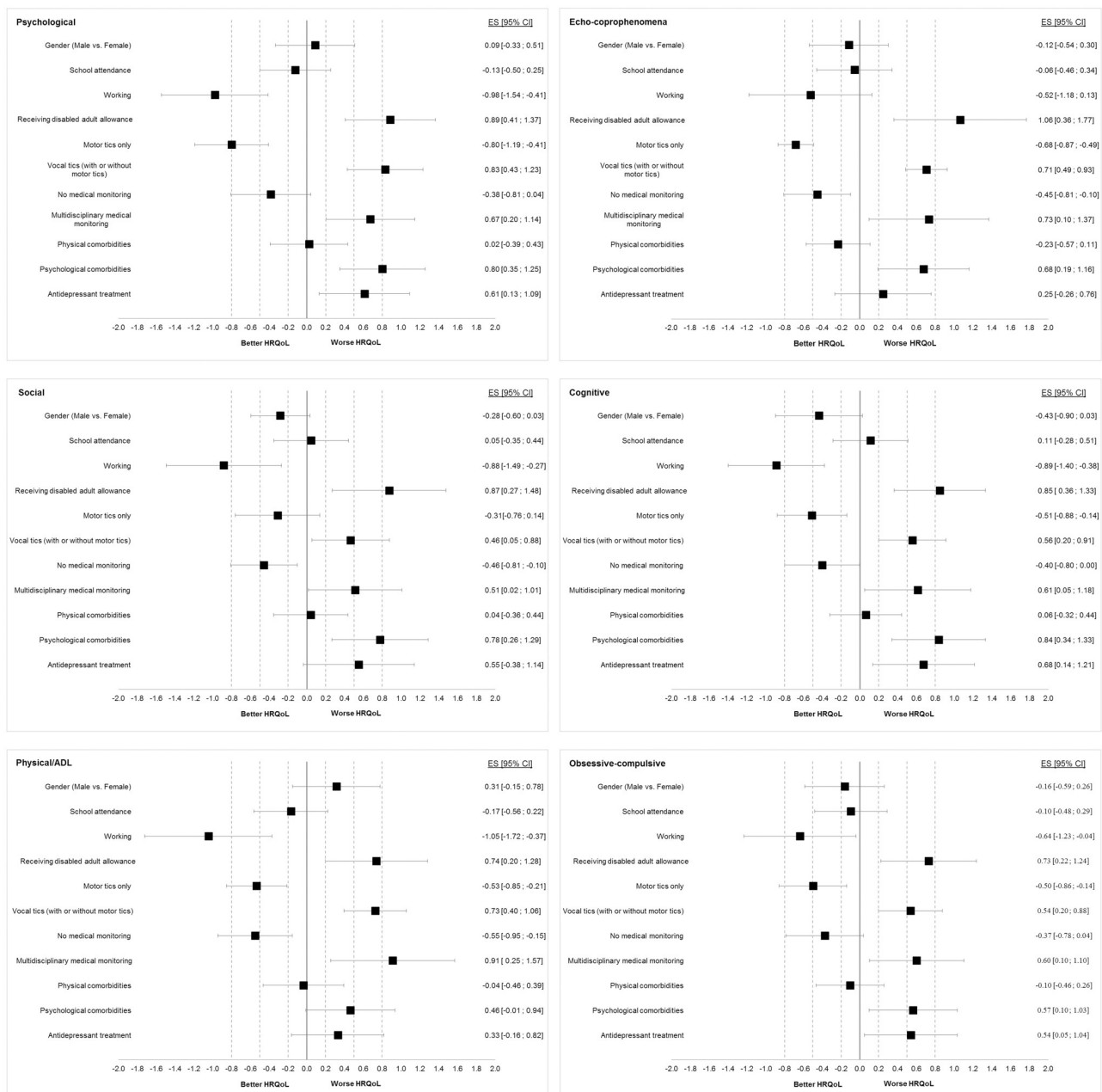

**Fig 1. Effect size of sociodemographic and clinical factors on GTS-QOL-French subscales.** Effect sizes (ES) are presented as forest plot with 95% confidence interval [95% CI] for each GTS-QOL-French subscale. Except for gender, the ES are given for 'Yes' versus 'No' values of the sociodemographic and clinical factors. Positive ES indicates worse health-related quality of life (HRQoL) and negative ES better HRQoL. Dotted lines represent the threshold for small (0.2), moderate (05) and large (0.8) ES.

with worse HRQoL, with moderate ES for the 'Psychological', 'Echo-coprophenomena', 'Social', 'Cognitive' and 'Obsessive-compulsive' subscales and with large ES for the 'Physical/ADL' subscale.

Having psychological comorbidities was associated with significantly worse HRQoL, with moderate ES for the 'Echo-coprophenomena', 'Social' and 'Obsessive-compulsive' subscales and large ES for the 'Psychological' and 'Cognitive' subscales.

Participants on antidepressants had a significantly poorer HRQoL for the 'Psychological', 'Cognitive' and 'Obsessive-compulsive' subscales, with moderate associated ES.

## Discussion

The present study describes the cross-cultural adaptation and validation of the GTS-QOL-French, including assessment of the convergent validity with the WHOQOL-BREF, which was not explored in the original UK study [11]. Our study is the first to use a disease-specific questionnaire to evaluate the HRQoL of a large sample of GTS subjects aged 16 years and over living in France.

HRQoL, taking into account the subjective point of view of subjects with GTS, contributes to a critical measure of the clinical outcome. A number of studies highlighted the phenotypic heterogeneity of GTS and the various course and prognostic. The GTS-QOL can help clinicians and researchers capture the impact of tics and other associated symptoms on patients' lives [36]. Therefore, the GTS-QOL-French offers these possibilities to French-speaking GTS subjects. The GTS-QOL could also allow the French-speaking caregivers of GTS population to be more directly informed about the HRQoL of their adolescents and adults. Additionally, it will enable investigators to propose French-speaking GTS subjects as participants in collaboration research projects using this scale.

Our GTS population can be considered as fully representative since it was similar to GTS populations described elsewhere in terms of male-female ratio and age at first symptoms [1, 11, 21, 37, 38]. Our GTS subjects were recruited from a Reference Center (tertiary referral center), a Competence Center (secondary referral center) and a patient association. Thus, they might be more representative of different levels of severity in GTS and of a wider community than subjects recruited solely in tertiary referral centers that receive patients with the most severe GTS. This is suggested when comparing the MOVES scores of the participants in our study to those in other studies [39–41]. Our GTS subjects also had a psychiatric comorbidity rate lower than that in studies that recruited persons exclusively in tertiary centers [40, 42]. This difference in rate could be due to the greater diversity of recruitment. However, although we took the precaution of estimating the rate from both patient-declared comorbidities and medical treatments, we cannot rule out that the rate was slightly underestimated. Likewise, we cannot rule out that subjects with GTS who had a greater number of tics and comorbidities, in particular psychiatric comorbidities, did not take part in the survey. Medical treatment of the participants was consistent with guidelines on pharmacotherapy in GTS including for behavioral disturbances [43, 44].

Our study, by assessing data completeness, provides more detailed results for respondent acceptability than in the original article. The GTS-QOL-French had good acceptability with excellent response rates and good response distribution, all of which indicate that the questionnaire was adapted to the population studied. However, 25 younger participants with a median age of 22.4 vs 29.0 years (p = 0.0250), who had higher MOVES scores [vocal tics (p = 0.0006), obsessions (p = 0.0250), compulsions (p = 0.0034), total score (p = 0.01)] and a more frequent multidisciplinary follow-up [45 vs 18% (p = 0.009)] took part in the study but declared having received help (reading or writing) to fill in the questionnaire.

The six-factor structure of the GTS-QOL-French differs from the four-factor structure of the original version of the GTS-QOL. The French version includes two additional subscales, 'Echo-coprophenomena' and 'Social', and the other four subscales were not exactly composed of the same items. The cross-culturally validated version of the GTS-QOL for children and adolescents (C&A-GTS-QOL), adapted from Italian to English, also identified six subscales, with additional 'Copro-phenomena' and 'Activities of daily living' subscales [45]. Neither the

original validation of GTS-QOL [11] nor the English and Italian C&A-GTS-QOL validations [14, 45] indicated whether items loaded on several subscales.

The 'Echo-coprophenomena' subscale groups together two items from the 'Physical/ADL' subscale and two items from the 'Obsessive-Compulsive' subscale of the original version. The item 'Difficulty seeing friends' grouped with the item 'Difficulty taking part in social activities' led to the establishment of the 'Social' subscale, which appears in the GTS-QOL-French. The item 'Loneliness/isolation' overlaps between factors 1 and 3 in the French version but we opted to place it in the 'Social' subscale, because in the resulting configuration, we observed better psychometric qualities. Unlike in the original GTS-QOL, the 'Psychological' subscale does not include the items 'Difficulty seeing friends' and 'Loneliness/isolation'. The item 'Phonic tics' was placed in the 'Physical/ADL' subscale because it is logical from a clinical point of view. This item evaluates vocal tics, but it does not evaluate the coprolalia and echolalia which are complex vocal tics including fully formed words with a semantically relevant content which are grouped with the complex motor tics which resemble intentional actions [36]. Moreover, in the resulting configuration, internal consistency of the 'Physical/ADL' subscale was better when it comprises item 4. Cronbach's alpha raised from 0.71 to 0.75 for the 'Physical/ADL' subscale if item 4 was added, and it remained almost constant for the 'Echo-coprophenomena' subscale if item 4 had been deleted (0.83 versus 0.84). Only the 'Cognitive' subscale remained identical between the original version and the GTS-QOL-French.

As in the original GTS-QOL study [11], we observed no ceiling effect. However, a clear-cut floor effect was observed for two subscales which did not exist in the original version. The floor effect of the 'Echo-coprophenomena' subscale was probably due to the lack of certain symptoms in our participants. The 'Other associated symptoms' MOVES score, such as copro- and echophenomena, were low in our participants compared to the scores reported in the study of Gaffney et al. (1994) [19] and in that which validated the original version of the GTS-QOL [11]. This could be explained by the low prevalence of coprophenomena and echophenomena in GTS subjects (4.9–25%) [46–48]. The floor effect of the 'Social' subscale is probably related to the absence of problems in the social sphere reported by a large number of participants in our study, who replied "satisfied" or "very satisfied" to items 20, 21 and 22 of the 'Social relationship' subscale of the WHOQOL-BREF (66%, 35% and 61%, respectively). Our more representative GTS population (not recruited solely in tertiary referral centers) could partly explain both. Additionally, the low prevalence of copro/echophenomena could partly explain the absence of problems in the social sphere reported by a large number of participants.

Internal consistency was good, which attests to the good homogeneity of the French version of GTS-QOL. The small to moderate correlations between the subscales imply that the GTS-QOL-French measures related but relatively different health constructs.

Lastly, the GTS-QOL-French had good convergent validity when compared to the MOVES questionnaire. However, because the original study does not give detailed results we cannot make a comparison of the two. Cavanna et al. (2008) compared the GTS-QOL with the Euro-Qol-5D (EQ-5D) [11]. The EQ-5D, is a very simple tool allowing for a very general assessment of quality of life and possesses only five items, with one item per dimension. It is mostly used in medico-economic studies [49]. We suggest a different approach that shows a good convergent validity of the GTS-QOL-French with the WHOQOL-BREF. The WHOQOL-BREF was used in a first study of the HRQoL of adults with GTS [8] and subsequently in a study of adolescents [50]. The 'Physical /ADL' subscale of the GTS-QOL-French groups together four items relating to involuntary movements, pain or physical injury due to tics, vocal tics and the impact on ADL and leisure activities. There are no common characteristics between the 'Environment' subscale of the WHOQOL-BREF and the 'Physical /ADL' and 'Obsessive/

compulsive' subscales of the GTS-QOL-French and consequently no significant correlation between them in our study. The 'Social relationships' subscale of the WHOQOL-BREF deals with personal relationships, sex life and the support the subject gets from his/her friends. Hence, there are no common characteristics between this subscale and the 'Physical/ADL' subscale of the GTS-QOL-French and therefore no significant correlation between them. As pointed out by Balsamo, Innamorati, & Lamis (2019) [51], we can underline the difficulty for integrating psychometrics and clinical views when inter-scales analysis allows us to go further and against apparent conclusions.

To the best of our knowledge, the present study is the first using a disease-specific questionnaire as primary outcome measure to give data on the HRQoL of GTS subjects living in France aged 16 years and above. The HRQoL of these participants was overall impaired, whether assessed by the GTS-QOL-French or WHOQOL-BREF. This finding confirms the results of the first study made in France with a generic scale [8] and those of other studies using the GTS-QOL in various European countries [11, 40–42, 52] or using generic scales in Europe and the USA [6, 7, 9].

The perception of HRQoL was mainly affected by psychological problems, followed by cognitive, physical and obsessive-compulsive aspects, as in the initial study [11] and in agreement with the meta-analysis of Evans et al. (2016) [3]. In addition, we show that participants with psychiatric comorbidities or antidepressant treatment had a poorer quality of life, with respectively large or moderate ES. These observations are evidence that psychological considerations (i.e. psychiatric/behavioral comorbidities or direct psychological consequences of the severity of GTS) are those that have the greatest influence on the HRQoL of GTS subjects. They are in line with those of the review of Evans et al. (2016) [3] and of previous studies of French people with GTS [8] and with the assessment of the effect of depression on EQ-5D scores [5]. Our study, complementary to the results of Eapen et al. (2016) [4], also measures the often significant impact of other factors, since the participants who were receiving a disabled adult allowance, who had vocal tics and multidisciplinary medical monitoring had worse HRQoL for all subscales.

Our study has certain limitations. Bias could have been introduced via the self-completion questionnaires because certain sociodemographic and medical data were supplied by the participants themselves. However, we took care to assess comorbidities both from those declared by the subjects and on the basis of the treatments they were receiving. For the reliability test, after exclusion of the respondents who had not maintained stable health status and treatment, or had experienced life events that could have disrupted their life between the two evaluations, 29 respondents were retained, a number close to the 28 subjects in the original study of Cavanna et al. (2008) [11], but lower than the 50 subjects recommended by Terwee et al. (2007) [16]. Nevertheless, we obtained ICC greater than the recommended 0.70. Finally, we did not study sensitivity to change after treatment.

## Conclusion

The GTS-QOL-French demonstrated satisfactory internal consistency, convergent validity and reliability. Further studies on responsiveness to change in longitudinal studies with therapeutic interventions are needed. The factor structure of the French version did not reflect the original structure of the GTS-QOL and further investigation on larger sample is required. Nevertheless, the cross-cultural adaptation of this specific instrument will now make it possible to assess with a disease-specific questionnaire the HRQoL of French-speaking GTS subjects aged 16 years and over.

## Supporting information

**S1 File. The GTS-QOL-French questionnaire.**
(PDF)

## Acknowledgments

Members of the The Syndrome de Gilles de La Tourette Study Group are Diane Cyrille, Philippe Derost, Loïc Duron, Candy Guiguet-Auclair, Isabelle Jalenques (leader of the group, ijalenques@chu-clermontferrand.fr), Sophie Lauron, Guillaume Legrand and Fabien Rondepierre (CHU Clermont-Ferrand, Clermont-Ferrand, France); Clara Jameux, Urbain Tauveron-Jalenques and Jeffrey Watts (Université Clermont Auvergne, Clermont-Ferrand, France); Andreas Hartmann (Assistance Publique-Hôpitaux de Paris AP-HP, Paris, France).

The authors thank the participants, the "*Association Française Syndrome Gilles de la Tourette*" and J Watts for advice on the English version of the manuscript.

## Author Contributions

**Conceptualization:** Isabelle Jalenques, Sophie Lauron.

**Data curation:** Candy Guiguet-Auclair, Fabien Rondepierre.

**Formal analysis:** Isabelle Jalenques, Diane Cyrille, Candy Guiguet-Auclair, Fabien Rondepierre.

**Funding acquisition:** Isabelle Jalenques, Fabien Rondepierre.

**Investigation:** Isabelle Jalenques, Diane Cyrille, Philippe Derost, Andreas Hartmann, Clara Jameux.

**Methodology:** Isabelle Jalenques, Fabien Rondepierre.

**Project administration:** Fabien Rondepierre.

**Resources:** Fabien Rondepierre.

**Software:** Candy Guiguet-Auclair.

**Supervision:** Isabelle Jalenques.

**Validation:** Isabelle Jalenques, Philippe Derost, Andreas Hartmann.

**Writing – original draft:** Isabelle Jalenques, Candy Guiguet-Auclair, Fabien Rondepierre.

**Writing – review & editing:** Isabelle Jalenques, Andreas Hartmann, Sophie Lauron, Urbain Tauveron-Jalenques, Candy Guiguet-Auclair.

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
