## [Decision Letter · Decision Letter 0]

30 Sep 2020

PONE-D-20-21404

The GTS-QOL (Gilles de la Tourette Syndrome–Quality of Life Scale): cross cultural evaluation of the French version

PLOS ONE

Dear Dr. Jalenques,

Thank you for submitting your manuscript to PLOS ONE. After careful consideration, we feel that it has merit but does not fully meet PLOS ONE’s publication criteria as it currently stands. Therefore, we invite you to submit a revised version of the manuscript that addresses the points raised during the review process.

We look forward to receiving your revised manuscript.

Kind regards,

Sandra Carvalho

Academic Editor

PLOS ONE

Journal Requirements:

3. One of the noted authors is a group; The Syndrome de Gilles de La Tourette Study Group.

In addition to naming the author group, please list the individual authors and affiliations within this group in the acknowledgments section of your manuscript.

Please also indicate clearly a lead author for this group along with a contact email address.

Reviewers' comments:

Reviewer's Responses to Questions

**Comments to the Author**

1. Is the manuscript technically sound, and do the data support the conclusions?

Reviewer #1: Yes

Reviewer #2: Partly

2. Has the statistical analysis been performed appropriately and rigorously? 

Reviewer #1: Yes

Reviewer #2: I Don't Know

3. Have the authors made all data underlying the findings in their manuscript fully available?

Reviewer #1: Yes

Reviewer #2: Yes

4. Is the manuscript presented in an intelligible fashion and written in standard English?

Reviewer #1: No

Reviewer #2: Yes

5. Review Comments to the Author

Reviewer #1: Review of: The GTS-QOL (Gilles de la Tourette Syndrome–Quality of Life Scale): cross-cultural evaluation of the French version

The document contains excellent work to illustrate the conditions for the GTS-QOL-French version. All protocols and analytical steps are relevant and necessary for future scientific and clinical consideration. My humble observations are just a few:

1. Why called it a “Cross-cultural” adaptation? I thought you maybe need a more unnecessary description of procedures, like other authors suggested before (i.e., Borsa, Damásio & Bandeira, 2012). Just called validation or cross-validation is enough.

2. Check all document for colloquial expressions (i.e., “very good acceptability”).

3. I thought a desirable description of the validation process should sound like the next paragraphs when your process fit it in (this a frame inspired by Cavanna et al., 2013):

Initially, the forward-backward procedure was applied to translate the GTS-QOL from English into ________ (Stage 1: scale translation). Two professional translators translated the questionnaire into _______ (“forward translation”). The resultant version was backwards translated into English (“backward translation”) by ____(number of) blind professional translators. The authors compared the two translated versions with the original English version, to yield the linguistic validation of the provisional questionnaire in ___________. In order to check the Italian population’s understanding and interpretation of the translated items, the questionnaire was pre-tested on ____(number of) patients with GTS (clinical sample). The results were discussed between experts and patients. This process led to a new _________ version of the GTS-QOL.

Secondly, this scale was adapted for adolescents and adults through the following steps (Stage 2: scale adaptation):

1. ____(number of) clinicians with expertise in the management of adolescents and adults with GTS independently suggested how to simplify and rephrase items that they considered to be confusing for _____, and how to change the context of items referring to adult life (e.g. job) to fit with an adolescent’s routine (i.e., high school life).

2. The experts’ opinions for each of the 27 items were compared and discussed with patients with GTS, and the questionnaire was modified accordingly. The GTS patients no express any remarkable difficult to understand any item.

3. The questionnaire was administered to ______ (number of) adolescents and ______ (number of) adults randomly selected from a nonclinical sample (school population). The recruited subjects were asked to comment on the comprehensibility of the items and to put them into their own words.

4. The same expert clinicians made wording adjustments.

5. The adjusted questionnaire was administered to ______ (number of) adolescents and ______ (number of) adults to identify any further confusing items or words. Items rated as confusing by ____% of the total sample were reworded or replaced.

Thirdly, the psychometric properties of the GTS-QOL have examined in an independent sample of ______ (number of) patients with GTS, described in Table _____.

4. Move “Table 1” to upper pages in the Participants sample section.

5. Participants should be described formally, sounding like this:

The final sample was recruited from three French specialist centres (GTS Reference Centre (Paris), GTS Competence Centre (Clermont- Ferrand), the “Association Française Syndrome Gilles de la Tourette”). Participants were aged 16–64 years and had no ______ (Type of) disabilities or other neurological conditions, and met DSM-IV-TR criteria for the diagnosis of GTS. Neuropsychiatrists evaluated all participants with substantial expertise in the management of GTS, who performed the neurological examination, clinical interview and cognitive evaluation.

6. There are no changes between the original GTS-QOL questionnaire and the French version (Could you name it something like GTS-QOL-French”)? In lines 160 to 176, you talked about two versions, the original and the French. This information is required before, not with the analysis conducted and reported in results, but in a straightforward way to the reader when described the applied instrument.

7. Please present first all instruments and them the Data collection information restricted to how the instruments were administrated.

8. Please, check your Results’ ordination and my suggestion:

Original document Suggestion (use it for results and discussion)

(i) Factor analysis. Data completeness.

(ii) Data completeness. (ii) Descriptive statistics and score distributions + Association between sociodemographic and clinical characteristics and GTS-QOL subscale scores (Lines 99 till 135).

(iii) Descriptive statistics and score distributions. (iii) Internal consistency: Cronbach’s α coefficient

(iv) Internal consistency: Cronbach’s α coefficient (iv) Inter-subscale correlations: Spearman’s coefficients

(v) Item-total correlations (v) Factor analysis.

(vi) Inter-subscale correlations: Spearman’s coefficients (vi) Item-total correlations

(vii) Convergent validity:TherelationshipsbetweenGTS-QOLand(1)WHOQOL-BREF,(2) MOVES subscales and (3) GTS-QOL VAS were studied, by calculating Spearman ρ correlation coefficients. (vii) Reliability

(viii) Reliability: Stability over time was assessed by the test-retest method. Reliability of the subscales was estimated by intraclass correlation coefficient (ICC), (viii) Convergent validity

9. Line 148, please pause with a period:

148. of tic severity in GTS than cases recruited solely in tertiary centres. As suggested by comparison …

10. Line 159, the subtitle is not necessary (or you need to include all other subtitles sections):

159. Cross-cultural evaluation of the French version of the GTS-QOL

11. Line 177, if you assume my suggestion for results’ order, might be the first Discussion’s section line.

12. Line 206 (p. 28), “Of note…” is colloquial.

13. Line 249 (p. 30), please check the English language.

14. About Discussion: Some lines are evident and repeat information reported in Results. Otherwise, some require a better explanation, as an example: “Reference Centre, a Competence Centre and a patient association and thus might be more representative of the different levels of tic severity in GTS than cases recruited solely in tertiary centres (Lines 146-148)” Why those centres have more representative population in France?

15. Please consider APA guidelines for citation in expressions like: “echo phenomena, were low in our participants compared to the scores reported in the study of Gaffney [18] and in that which validated the original version of the GTS-QOL [11]. (Lines 188-189)”.

16. Lines 208-220 contain an explanation for no significant correlation between WHOQOL-BREF subscales and some subscales of the GTS-QOL. The explanation indicates “There are no common characteristics” between not correlated subscales. I know that it is not an objective for the study. However, following Balsamo, Innamorati, & Lamis (2019), you might comment about the difficulty for integrating psychometrics and clinical views when inter-scales analysis allows us to go further and against apparent conclusions.

17. Why there are not any health or well-being policy recommendations about GTS population and benefits for using GTS-QOL-French (i.e., French-speaking caregivers of GTS population more directly informed about the life-quality of their adolescents and adults?

References

Balsamo, M., Innamorati, M., & Lamis, D. A. (2019). Editorial: Clinical Psychometrics: Old Issues and New Perspectives. Frontiers in psychology, 10, 947. https://doi.org/10.3389/fpsyg.2019.00947

Cavanna, A. E., Luoni, C., Selvini, C., Blangiardo, R., Eddy, C. M., Silvestri, P. R., Calì, P. V., Seri, S., Balottin, U., Cardona, F., Rizzo, R., & Termine, C. (2013). The Gilles de la Tourette Syndrome-Quality of Life Scale for children and adolescents (C&A-GTS-QOL): development and validation of the Italian version. Behavioural neurology, 27(1), 95–103. https://doi.org/10.3233/BEN-120274

Reviewer #2: Summary:

In this manuscript [“The GTS-QOL (Gilles de la Tourette Syndrome–Quality of Life Scale): cross cultural evaluation of the French version”], Jalenques et al. present the methodology and results underlying the development of a French version of the GTS-QOL. The analyzed sample (109 participants aged 16-years old or above) and applied analyzes seem appropriate, and the results are overall quite satisfying; thus, I see no reason for this manuscript not to be accepted in PLOS ONE, provided that, at least, the following comments/doubts are addressed. (I hope that you find them helpful, and please note that some of my comments/doubts might apply to several parts of the manuscript; not only to the quoted portions that I pasted below.)

Main comments:

The manuscript is very well written and easy to follow; thus, I would like to start by congratulating the authors. In its current state, however, the Discussion of the manuscript seems a bit poor. Several portions of text are too descriptive, lacking a more comprehensive interpretation of the results and, perhaps more importantly, of the pros and cons of the French version of the GTS-QOL presented here. Several flaws of the developed version are indicated in the “Results” section, namely the abnormal loading of some items and the considerable floor effects for some of its subscales. The latter, for instance, seems very problematic if one desires to apply this scale to patients with lower symptom severity, meaning that, in the future, researchers that wish to use this scale will necessarily have to use carefully designed recruitment strategies... Otherwise, this version might not be suitable.

Although I am not an expert in the GTS-QOL, I am aware of, at least, an Italian version of this questionnaire. Even though such translated version was applied to children and adolescents (which complicates the interpretation of its properties in comparison to the version here presented), I think that it would greatly improve the paper to discuss the results from the French version in comparison not only to the original version but also in comparison to other translated versions of the scale. If possible, the focus should be on versions validated in adults, but, if that is not possible, I think that even the analysis of the versions validated in children and adolescents might provide valuable information to a reader that desires to better understand the pros and cons of this version.

Main doubt:

Should not the translated version of the scale be made available with the article?

Other (minor) comments/doubts:

1. Mentioning “very good acceptability” in the Abstract, providing no quantitative measure of such acceptability, seems suboptimal.

2. Using “ICC” without introducing the acronym in the Abstract seems suboptimal too, but I am guessing that might relate to the maximum number of characters allowed in the Abstract…

3. The choice of the MOVES and WHOQOL-BREF scales should be more comprehensively explained so that it is easier for the readers to follow the authors rationale from the beginning.

4. It would be nice to incorporate some quantitative information into this following sentence: “It had good acceptability, validity, internal consistency and test-retest reliability”.

5. The rationale underlying the selection of the exact subsample that was used to assess the scale’s reliability should be much better explained.

6. Are the “quality criteria” from references 13 and 14 independent from the specific properties of the questionnaire? It would make sense that they were dependent on the total number of questions, number of questions per subscale, etc. Can you please clarify?

7. In the “The GTS-QOL questionnaire” subsection of the Methods, the number of items per subscale should be mentioned.

8. “Good understanding and content validity was shown for all items.”: Can you please explain how was this “shown”?

9. “The generic WHOQOL-BREF questionnaire comprised 26 items which are grouped into four

subscales: ‘Physical health’, ‘Psychological health’, ‘Social relationships’ and ‘Environment’.

For each subscale, scores are rated between 0 and 100, with higher values indicating better

HRQoL”: Should maybe emphasize that this is contrary to GTS-QOL and indicate how many items per subscale in the sentence above.

10. Some references seem to be missing in: “Factor analysis: Factor analysis with the principal axis extraction method and oblique promax rotation were performed to study the multidimensionality and distribution of the items in subscales.”

11. The authors mention that “Eigenvalues higher than 1 (Kaiser criterion) and Cattell’s scree plot [24] were used to verify factor solution accuracy”, but then fail to address exactly how they did so.

12. Given that tests were two-tailed, the use of “attempted” rather than, e.g., “expected”, in page 14, seems suboptimal.

13. It was not clear for me what were the independent groups in “Finally, comparisons of GTS-QOL subscale scores between independent groups (…)”.

14. Not sure that the use of “more” in the beginning of page 15 is correct. Should not it be “lower”?

15. Table 1 seems to have some errors. Some of the categories should add to 100% and that is not the case. In some categories that might have been simply due to rounding errors (e.g., “Education”) while in others (e.g., “Comorbidities”) that does not seem to be the case.

16. In Table 2, I think that it would be helpful to indicate the maximum possible values too, to facilitate the interpretation of the presented values.

17. The rationale for keeping item 4 in factor 5 was not clear; it seemed a somewhat odd choice, especially given the relation between phonic tics and some of the items in the “Echo-coprophenomena” scale.

18. The designation of “Cognitive” seems somewhat misleading, given the nature of the items in that scale, but I guess that might be due to the original English version…

19. In Table 5, the problematic items (9, 10, and possibly 15 too) from the “Obsessive-compulsive” scale should be highlighted somehow, so that the readers can identify them more easily.

20. In page 20, I think it would be better to specify some of the values instead of only mentioning “small” or “moderate” correlations.

21. Are all p-values in Table 7 correct? It would seem more logical if they were 0.01 and 0.001 instead, given that * corresponds to 0.05.

22. Can you please explain if the EuroQol-5D has important flaws that one should be aware? That would help understanding why you suggested a different approach.

23. I do not quite agree with the following sentence: “These observations are evidence that psychological 234 considerations are those that have the greatest influence on the HRQoL of GTS subjects”. Psychological symptoms might themselves be a direct consequence of the severity of Tourette syndrome and/or comorbidities symptoms…

Typos:

1. There is an extra “)” following “[15]” in the “Translation and cultural adaptation of the GTS-QOL” subsection.

2. There is a blank space missing before “[28]”.

6. PLOS authors have the option to publish the peer review history of their article (what does this mean?). If published, this will include your full peer review and any attached files.

Reviewer #1: **Yes: **Juan J Giraldo-Huertas

Department of Development and Education Psychology

Universidad de la Sabana

Reviewer #2: No

---

## [Author Response · Author response to Decision Letter 0]

13 Nov 2020

Response to reviewers

Comments to the Author

1. Is the manuscript technically sound, and do the data support the conclusions?

Reviewer #1: Yes

Reviewer #2: Partly

The discussion and the conclusion have been partly rewritten.

2. Has the statistical analysis been performed appropriately and rigorously? 

Reviewer #1: Yes

Reviewer #2: I Don't Know

3. Have the authors made all data underlying the findings in their manuscript fully available?

Reviewer #1: Yes

Reviewer #2: Yes

4. Is the manuscript presented in an intelligible fashion and written in standard English?

Reviewer #1: No

Reviewer #2: Yes

The manuscript has been reviewed by a native English speaker.

 

5. Review Comments to the Author

Reviewer #1: Review of: The GTS-QOL (Gilles de la Tourette Syndrome–Quality of Life Scale): cross-cultural evaluation of the French version

The document contains excellent work to illustrate the conditions for the GTS-QOL-French version. All protocols and analytical steps are relevant and necessary for future scientific and clinical consideration. My humble observations are just a few:

1. Why called it a “Cross-cultural” adaptation? I thought you maybe need a more unnecessary description of procedures, like other authors suggested before (i.e., Borsa, Damásio & Bandeira, 2012). Just called validation or cross-validation is enough.

We did a careful bibliographic search in many journals including Plos One, and a very large number of studies on translation and validation of scales are titled “cross-cultural adaptation and validation/evaluation”. We therefore decided to modify the title of our work to:

“Cross-cultural adaptation and psychometric evaluation of the French version of the Gilles de la Tourette Syndrome Quality of Life Scale (GTS-QOL) ».

This is in accordance with the procedure described in Borsa, Damásio & Bandeira, 2012. We have adapted this modification throughout the manuscript, especially in the method section.

2. Check all document for colloquial expressions (i.e., “very good acceptability”).

The manuscript has been reviewed by a native English speaker and colloquial expressions have been removed.

3. I thought a desirable description of the validation process should sound like the next paragraphs when your process fit it in (this a frame inspired by Cavanna et al., 2013):

Initially, the forward-backward procedure was applied to translate the GTS-QOL from English into ________ (Stage 1: scale translation). Two professional translators translated the questionnaire into _______ (“forward translation”). The resultant version was backwards translated into English (“backward translation”) by ____(number of) blind professional translators. The authors compared the two translated versions with the original English version, to yield the linguistic validation of the provisional questionnaire in ___________. In order to check the Italian population’s understanding and interpretation of the translated items, the questionnaire was pre-tested on ____(number of) patients with GTS (clinical sample). The results were discussed between experts and patients. This process led to a new _________ version of the GTS-QOL.

Secondly, this scale was adapted for adolescents and adults through the following steps (Stage 2: scale adaptation):

1. ____(number of) clinicians with expertise in the management of adolescents and adults with GTS independently suggested how to simplify and rephrase items that they considered to be confusing for _____, and how to change the context of items referring to adult life (e.g. job) to fit with an adolescent’s routine (i.e., high school life).

2. The experts’ opinions for each of the 27 items were compared and discussed with patients with GTS, and the questionnaire was modified accordingly. The GTS patients no express any remarkable difficult to understand any item.

3. The questionnaire was administered to ______ (number of) adolescents and ______ (number of) adults randomly selected from a nonclinical sample (school population). The recruited subjects were asked to comment on the comprehensibility of the items and to put them into their own words.

4. The same expert clinicians made wording adjustments.

5. The adjusted questionnaire was administered to ______ (number of) adolescents and ______ (number of) adults to identify any further confusing items or words. Items rated as confusing by ____% of the total sample were reworded or replaced.

Thirdly, the psychometric properties of the GTS-QOL have examined in an independent sample of ______ (number of) patients with GTS, described in Table _____.

The method section has been rewritten as recommended (page 4-5):

 “Phase 1 - Translation and cross-cultural adaptation of the French version of the GTS-QOL

The GTS-QOL was adapted from English into French following established international guidelines [13,14]. First, the forward-backward procedure was applied to translate the GTS-QOL from English to French (Stage 1: scale translation). Three informed bilingual translators (2 psychiatrists and 1 neurologist) experienced in GTS research and one bilingual translator naive to the outcome measure independently translated the questionnaire into French (“forward translation”). All had French as their mother tongue and are fluent in English. The resultant version was translated back to English (“backward translation”) by a native English professional translator fluent in French, blinded to the original English version and with a medical background. The authors compared the translated versions with the original English version to yield the linguistic validation of the provisional questionnaire in French. They discussed item-translation, semantic, idiomatic, experiential, and conceptual equivalents. In order to check the French-speaking adult’s understanding and interpretation of the translated items, the questionnaire was pre-tested on 10 adults with GTS (clinical sample). The results were discussed between experts and patients.

Secondly, this scale was adapted for adolescents (Stage 2: scale adaptation) by three clinicians with expertise in the management of adolescents with GTS. They suggested how to simplify and rephrase items that they considered to be confusing for adolescents. The questionnaire was administered to 10 adolescents. They were asked to comment on the comprehensibility of the items. Their responses and comments were reviewed. The same experts did not suggest any wording adjustments. The French version was tested on a sample of GTS subjects (4 adults and 3 adolescents) to evaluate the comprehensibility of instructions and items. No understanding difficulty was noticed. Consequently, the expert committee adopted this version as the pre-final cross-cultural adaptation (supplementary data). We named this version the GTS-QOL-French.

Thirdly, the psychometric properties of the GTS-QOL-French were examined in an independent sample of 109 people with GTS (80 males; age range: 16-64 years) (Stage 3: scale evaluation).”

4. Move “Table 1” to upper pages in the Participants sample section.

The table 1 has been moved in the participants section as recommended.

5. Participants should be described formally, sounding like this:

The final sample was recruited from three French specialist centres (GTS Reference Centre (Paris), GTS Competence Centre (Clermont- Ferrand), the “Association Française Syndrome Gilles de la Tourette”). Participants were aged 16–64 years and had no ______ (Type of) disabilities or other neurological conditions, and met DSM-IV-TR criteria for the diagnosis of GTS. Neuropsychiatrists evaluated all participants with substantial expertise in the management of GTS, who performed the neurological examination, clinical interview and cognitive evaluation.

The description of the participants has been rewritten as recommended (Page 6):

“The sample was recruited from two French specialist centres [GTS Reference Centre (Paris), GTS Competence Centre (Clermont- Ferrand)] and the “Association Française Syndrome Gilles de la Tourette”. Participants were aged 16 years and above. They met DSM-IV-TR criteria for the diagnosis of GTS and were able to complete the questionnaires without help. A neurologist with substantial expertise in the management of GTS evaluated all participants and performed the neurological examination. The GTS subjects with a postal address who consented to participate then received a set of questionnaires to self-complete.”

6. There are no changes between the original GTS-QOL questionnaire and the French version (Could you name it something like GTS-QOL-French”)? 

The French version of the GTS-QOL has been named GTS-QOL-French in all the manuscript.

In lines 160 to 176, you talked about two versions, the original and the French. This information is required before, not with the analysis conducted and reported in results, but in a straightforward way to the reader when described the applied instrument.

The description of the original version of the GTS-QOL was added in the introduction. At the end of the paragraph presenting the translation and cross-cultural adaptation in the method section, we indicated that the translated version was named GTS-QOL-French.

7. Please present first all instruments and them the Data collection information restricted to how the instruments were administrated.

The paragraph was called “study variables and instruments” instead of “data collection” in order to avoid confusion (pages 9-10).

8. Please, check your Results’ ordination and my suggestion:

Original document Suggestion (use it for results and discussion)

(i) Factor analysis. 

(ii) Data completeness. 

(iii) Descriptive statistics and score distributions. 

(iv) Internal consistency: Cronbach’s α coefficient 

(v) Item-total correlations 

(vi) Inter-subscale correlations: Spearman’s coefficients

(vii) Convergent validity:The relationships between GTS-QOL and(1)WHOQOL-BREF,(2) MOVES subscales and (3) GTS-QOL VAS were studied, by calculating Spearman ρ correlation coefficients.

(viii) Reliability: Stability over time was assessed by the test-retest method. Reliability of the subscales was estimated by intraclass correlation coefficient (ICC),

(i) Data completeness.

(ii) Descriptive statistics and score distributions + Association between sociodemographic and clinical characteristics and GTS-QOL subscale scores (Lines 99 till 135).

(iii) Internal consistency: Cronbach’s α coefficient

(iv) Inter-subscale correlations: Spearman’s coefficients

(v) Factor analysis.

 (vi) Item-total correlations

 (vii) Reliability

 (viii) Convergent validity

We understand your proposition, this order of presentation of results being usual in the evaluation of psychometric properties. Nevertheless, we maintain the fact that the results of the factor analysis must be presented at the beginning of the section. Indeed, the 4-factor structure of the original version is not found in the French version. Our analyzes demonstrate a structure in 6 factors that we have named. The following analyzes (descriptive statistics and score distributions, internal consistency, reliability, convergent validity) focus on these new dimensions (with their names). It therefore seems important to us to present their origin before using them in other analyzes.

Su et al. presented the results of the psychometric validation of the English version of the C & A-GTS-QOL in the same order, beginning with the factor analysis. They also found a different factorial structure in their English version compared to the original Italian version.

Reference: Su MT, McFarlane F, Cavanna AE, Termine C, Murray I, Heidemeyer L, et al. The English Version of the Gilles de la Tourette Syndrome-Quality of Life Scale for Children and Adolescents (C&A-GTS-QOL). J Child Neurol. 2017;32:76‑83.

9. Line 148, please pause with a period:

P32

Our GTS subjects were recruited from a Reference Center (tertiary referral center), a Competence Center (secondary referral center) and a patient association. Thus, they might be more representative of different levels of severity in GTS and of a wider community than subjects recruited solely in tertiary referral centers that receive patients with the most severe GTS. This is suggested when comparing the MOVES scores of the participants in our study to those in other studies [39–41].

10. Line 159, the subtitle is not necessary (or you need to include all other subtitles sections):

159. Cross-cultural evaluation of the French version of the GTS-QOL

We removed the subtitle in the discussion.

11. Line 177, if you assume my suggestion for results’ order, might be the first Discussion’s section line.

The order of certain points of discussion has been partially revised according to your remarks. 

12. Line 206 (p. 28), “Of note…” is colloquial.

All colloquial expressions were removed as page 35:

“Of note, The WHOQOL-BREF was used in a first study of the HRQoL of adults with GTS [8] and subsequently in a study of adolescents [50].”

13. Line 249 (p. 30), please check the English language.

The language was checked and changes have been made (p36).

“…. 29 respondents were retained, a number close to that of the 28 subjects in the original study of Cavanna et al. (2008) [11], but lower than that of the 50 subjects recommended by Terwee et al. (2007) [16].”

14. About Discussion: Some lines are evident and repeat information reported in Results. Otherwise, some require a better explanation, as an example: “Reference Centre, a Competence Centre and a patient association and thus might be more representative of the different levels of tic severity in GTS than cases recruited solely in tertiary centres (Lines 146-148)” Why those centres have more representative population in France?

Repeated information has been removed from the discussion.

Other information has been explained more fully, as in page 32:

“Our GTS subjects were recruited from a Reference Center (tertiary referral center), a Competence Center (secondary referral center) and a patient association. Thus, they might be more representative of different levels of severity in GTS and of a wider community than subjects recruited solely in tertiary referral centers that receive patients with the most severe GTS. This is suggested when comparing the MOVES scores of the participants in our study to those in other studies [39–41].”

15. Please consider APA guidelines for citation in expressions like: “echo phenomena, were low in our participants compared to the scores reported in the study of Gaffney [18] and in that which validated the original version of the GTS-QOL [11]. (Lines 188-189)”.

These kind of citations have been checked and adapted to APA guidelines.

The ‘Other associated symptoms’ MOVES score, such as copro- and echophenomena, were low in our participants compared to the scores reported in the study of Gaffney et al. (1994) [18] and in that which validated the original version of the GTS-QOL [11].

16. Lines 208-220 contain an explanation for no significant correlation between WHOQOL-BREF subscales and some subscales of the GTS-QOL. The explanation indicates “There are no common characteristics” between not correlated subscales. I know that it is not an objective for the study. However, following Balsamo, Innamorati, & Lamis (2019), you might comment about the difficulty for integrating psychometrics and clinical views when inter-scales analysis allows us to go further and against apparent conclusions.

We agree with your remark and we introduced it (p 35):

“As pointed out by Balsamo, Innamorati, & Lamis (2019) [51], we can underline the difficulty for integrating psychometrics and clinical views when inter-scales analysis allows us to go further and against apparent conclusions”

17. Why there are not any health or well-being policy recommendations about GTS population and benefits for using GTS-QOL-French (i.e., French-speaking caregivers of GTS population more directly informed about the life-quality of their adolescents and adults?

We reworked the discussion and introduced this notion (p 31-32):

“HRQoL, taking into account the subjective point of view of subjects with GTS, contributes to a critical measure of the clinical outcome. A number of studies highlighted the phenotypic heterogeneity of GTS and the various course and prognostic. The GTS-QOL can help clinicians and researchers capture the impact of tics and other associated symptoms on patients’ lives [36]. Therefore, the GTS-QOL-French offers these possibilities to French-speaking GTS subjects. The GTS-QOL could also allow the French-speaking caregivers of GTS population to be more directly informed about the HRQoL of their adolescents and adults. Additionally, it will enable investigators to propose French-speaking GTS subjects as participants in collaboration research projects using this scale.”

References

Balsamo, M., Innamorati, M., & Lamis, D. A. (2019). Editorial: Clinical Psychometrics: Old Issues and New Perspectives. Frontiers in psychology, 10, 947. https://doi.org/10.3389/fpsyg.2019.00947

Cavanna, A. E., Luoni, C., Selvini, C., Blangiardo, R., Eddy, C. M., Silvestri, P. R., Calì, P. V., Seri, S., Balottin, U., Cardona, F., Rizzo, R., & Termine, C. (2013). The Gilles de la Tourette Syndrome-Quality of Life Scale for children and adolescents (C&A-GTS-QOL): development and validation of the Italian version. Behavioural neurology, 27(1), 95–103. https://doi.org/10.3233/BEN-120274

 

Reviewer #2: Summary:

In this manuscript [“The GTS-QOL (Gilles de la Tourette Syndrome–Quality of Life Scale): cross cultural evaluation of the French version”], Jalenques et al. present the methodology and results underlying the development of a French version of the GTS-QOL. The analyzed sample (109 participants aged 16-years old or above) and applied analyzes seem appropriate, and the results are overall quite satisfying; thus, I see no reason for this manuscript not to be accepted in PLOS ONE, provided that, at least, the following comments/doubts are addressed. (I hope that you find them helpful, and please note that some of my comments/doubts might apply to several parts of the manuscript; not only to the quoted portions that I pasted below.)

Main comments:

The manuscript is very well written and easy to follow; thus, I would like to start by congratulating the authors. In its current state, however, the Discussion of the manuscript seems a bit poor. Several portions of text are too descriptive, lacking a more comprehensive interpretation of the results and, perhaps more importantly, of the pros and cons of the French version of the GTS-QOL presented here. Several flaws of the developed version are indicated in the “Results” section, namely the abnormal loading of some items and the considerable floor effects for some of its subscales. The latter, for instance, seems very problematic if one desires to apply this scale to patients with lower symptom severity, meaning that, in the future, researchers that wish to use this scale will necessarily have to use carefully designed recruitment strategies... Otherwise, this version might not be suitable.

The discussion has been rewritten and repeated information (from the results part)/descriptive sections have been removed. Abnormal loading of some items and floor effects where developed in the new discussion.

Concerning the floor effect of the ‘Echo-coprophenomena’ and ‘Social’ subscales, it could reflect our recruitment since participants were not only recruited from specific referral centers for GTS but also from an association. Participants reported low prevalence of coprophenomena and echophenomena (MOVES) and reported no problems in the social sphere (WHOQOL-BREF). Recruitment with participants coming only from specific referral centers would certainly have eliminated this floor effect, but the participants would have been less representative of adults with GTS (whose symptoms may be lower). 

Although I am not an expert in the GTS-QOL, I am aware of, at least, an Italian version of this questionnaire. Even though such translated version was applied to children and adolescents (which complicates the interpretation of its properties in comparison to the version here presented), I think that it would greatly improve the paper to discuss the results from the French version in comparison not only to the original version but also in comparison to other translated versions of the scale. If possible, the focus should be on versions validated in adults, but, if that is not possible, I think that even the analysis of the versions validated in children and adolescents might provide valuable information to a reader that desires to better understand the pros and cons of this version.

At our knowledge, there is no other translation of the adult version. A translated version (from Italian to English) exists for children and adolescents and showed a structure of 6 subscales as us. The two additional subscales were ‘Copro-phenomena’ and ‘Activities of daily living’. This information has been added in the discussion.

Main doubt:

Should not the translated version of the scale be made available with the article?

The GTS-QOL-French questionnaire has been added in supplementary data.

Other (minor) comments/doubts:

1. Mentioning “very good acceptability” in the Abstract, providing no quantitative measure of such acceptability, seems suboptimal.

Values have been added in the abstract

Results: The results showed good acceptability (missing values per subscale ranging from 0% to 0.9%)

2. Using “ICC” without introducing the acronym in the Abstract seems suboptimal too, but I am guessing that might relate to the maximum number of characters allowed in the Abstract…

The acronym of “ICC” has been added in the abstract

Results: 

good test–retest reliability (intraclass correlation coefficients ranging from 0.70 to 0.81). Concurrent validity with the MOVES and WHOQOL-BREF scales showed high correlations.

3. The choice of the MOVES and WHOQOL-BREF scales should be more comprehensively explained so that it is easier for the readers to follow the authors rationale from the beginning.

The “Instruments” section has been rewritten (page 10):

“The generic WHOQOL-BREF questionnaire comprised 26 items of which two concern overall perception of HRQoL and health, while the remaining 24 items are grouped into four subscales: ‘Physical health’ (7 items), ‘Psychological health’ (6 items), ‘Social relationships’ (3 items), and ‘Environment’ (8 items) [17]. For each subscale, scores are rated between 0 and 100, with higher values indicating better HRQoL, contrary to GTS-QOL. The WHOQOL-BREF is a cross-culturally valid assessment of HRQoL, with good to excellent reliability and validity in neuropsychiatric patients [17]. Its French version [18] has already been used very satisfactorily during a previous study in French-speaking adults with GTS [8].

The MOVES is a self-report scale measuring the severity of tics and other phenomena observed in GTS [19]. It comprises 20 items grouped into five subscales: ‘Motor tics’, ‘Vocal tics’, ‘Obsessions’, ‘Compulsions’ and ‘Other associated symptoms’ such as copro-, pali- and echophenomena. For each subscale, a score is obtained by adding the scores of the items listed in the subscale. A total score is calculated by adding the scores of the five subscales, which range from 0 (no symptom) to 60 (the worst condition). For clinical scoring, the ‘Motor tics’ and ‘Vocal tics’ scores are added to obtain a ‘Tics’ subscale score. The ‘Obsessions’ and ‘Compulsions’ scores are added to form an ‘Obsessive-compulsive’ subscale score. The MOVES is suggested as a severity scale for tics and related sensory phenomena and recommended as a screening instrument by the Committee on Rating Scale Development of the International Parkinson’s Disease and Movement Disorder Society [20]. It has been used by Cavanna et al [11] to evaluate the GTS-QOL. A previous study in French-speaking persons with GTS aged 12 years and over provided evidence of the good psychometric properties of the French version of the MOVES [21].”

4. It would be nice to incorporate some quantitative information into this following sentence: “It had good acceptability, validity, internal consistency and test-retest reliability”.

Quantitative informations have been added in this sentence as requested:

“It had good acceptability, validity (inter-subscales correlations ranging from 0.5 to 0.7), internal consistency (Cronbach’s alpha ≥ 0.8) and test-retest reliability (intraclass correlation coefficients – ICCs ≥ 0.8).”

5. The rationale underlying the selection of the exact subsample that was used to assess the scale’s reliability should be much better explained. 

To assess the test-retest reliability of the GTS-QOL-French, the questionnaire was mailed a second time to a subsample of participants (simple random sampling using random number tables) 15 days after the first assessment.

6. Are the “quality criteria” from references 13 and 14 independent from the specific properties of the questionnaire? It would make sense that they were dependent on the total number of questions, number of questions per subscale, etc. Can you please clarify? 

The quality criteria developed by Terwee et al. or reported in the COSMIN checklist are independent of the number of items per subscale and independent of the number of items in the questionnaire itself. 

The COSMIN checklist for example contains standards referring to design requirements and preferred statistical methods of studies on measurement properties, whatever the instrument evaluated. 

7. In the “The GTS-QOL questionnaire” subsection of the Methods, the number of items per subscale should be mentioned.

In the new version of the manuscript, the description of the original GTS-QOL has been moved to the introduction (page 3) and the number of items per subscale has been added:

“The original GTS-QOL comprises 27 items referring to the past 4 weeks and rated on a 5-point Likert scale, from 0 (no problem) to 4 (extreme problem), grouped into four subscales: ‘Psychological’ (11 items), ‘Physical and activities of daily living’ (7 items), ‘Obsessive-compulsive’ (5 items), and ‘Cognitive’ (4 items).”

8. “Good understanding and content validity was shown for all items.”: Can you please explain how was this “shown”? 

The same experts did not suggest any wording adjustments. The French version was tested on a sample of GTS subjects, 4 adults and 3 adolescents, to evaluate the comprehensibility of instructions and items. No understanding difficulty was noticed.

9. “The generic WHOQOL-BREF questionnaire comprised 26 items which are grouped into four subscales: ‘Physical health’, ‘Psychological health’, ‘Social relationships’ and ‘Environment’. For each subscale, scores are rated between 0 and 100, with higher values indicating better HRQoL”: Should maybe emphasize that this is contrary to GTS-QOL and indicate how many items per subscale in the sentence above.

These modifications have been made as recommended page 10):

“The generic WHOQOL-BREF questionnaire comprised 26 items of which two concern overall perception of QoL and health, while the remaining 24 items are grouped into four subscales: ‘Physical health’ (7 items), ‘Psychological health’ (6 items), ‘Social relationships’ (3 items), and ‘Environment’ (8 items). For each subscale, scores are rated between 0 and 100, with higher values indicating better HRQoL contrary to GTS-QOL.”

10. Some references seem to be missing in: “Factor analysis: Factor analysis with the principal axis extraction method and oblique promax rotation were performed to study the multidimensionality and distribution of the items in subscales.”

The reference of Costello and Osborne has been added: Costello AB, Osborne JW. Best practices in exploratory factor analysis: Four recommendations for getting the most from your analysis. Practical Assessment, Research & Evaluation. 2005;10(7).

11. The authors mention that “Eigenvalues higher than 1 (Kaiser criterion) and Cattell’s scree plot [24] were used to verify factor solution accuracy”, but then fail to address exactly how they did so.

In the method section, there was a mistake, “used to verify” was replaced by “Used” :

“Eigenvalues higher than 1 (Kaiser criterion) and Cattell’s scree plot [28] were used for factor retention”

In the results, a sentence was added:

“The factor analysis (promax rotation) of the 27 items of the GTS-QOL-French identified six factors with eigenvalues higher than one and accounting for 69.9% of the total variance (Table 3).”

12. Given that tests were two-tailed, the use of “attempted” rather than, e.g., “expected”, in page 14, seems suboptimal.

Changes have been made as advised (page 12):

“The relationships between GTS-QOL-French and (1) WHOQOL-BREF, (2) MOVES subscales and (3) GTS-QOL-French VAS were studied, by calculating Spearman ρ correlation coefficients. Negative correlations were expected between GTQ-QOL-French subscales, WHOQOL-BREF subscales, and GTS-QOL-French VAS since low scores indicated good conditions for the GST-QOL-French but bad conditions for the WHOQOL-BREF and GTS-QOL-French VAS. Positive correlations were expected between GTQ-QOL-French and MOVES subscales because low scores indicated good conditions for the two questionnaires.”

13. It was not clear for me what were the independent groups in “Finally, comparisons of GTS-QOL subscale scores between independent groups (…)”.

The GTS-QOL-French subscale scores were compared according to age, gender, disease duration, school attendance, professional activity, financial aid, motor and/or vocal tics, medical monitoring, comorbidities, treatment.

14. Not sure that the use of “more” in the beginning of page 15 is correct. Should not it be “lower”?

This sentence has been corrected:

“For all subscales of GTS-QOL-French, a high positive ES indicated an important negative impact on HRQoL.”

15. Table 1 seems to have some errors. Some of the categories should add to 100% and that is not the case. In some categories that might have been simply due to rounding errors (e.g., “Education”) while in others (e.g., “Comorbidities”) that does not seem to be the case.

Errors have been corrected and we added a table footnote to clarify that a participant may have different locations for first tics, different medical monitoring, and different treatments for GTS.

16. In Table 2, I think that it would be helpful to indicate the maximum possible values too, to facilitate the interpretation of the presented values. 

The possible minimum and maximum values have been added in table 2.

17. The rationale for keeping item 4 in factor 5 was not clear; it seemed a somewhat odd choice, especially given the relation between phonic tics and some of the items in the “Echo-coprophenomena” scale.

We clarified this choice in the discussion (page 33):

“The item ‘Phonic tics’ was placed in the ‘Physical/ADL’ subscale because it is logical from a clinical point of view. This item evaluates vocal tics, but it does not evaluate the coprolalia and echolalia which are complex vocal tics including fully formed words with a semantically relevant content which are grouped with the complex motor tics which resemble intentional actions [36]. Moreover, in the resulting configuration, internal consistency of the ‘Physical/ADL’ subscale was better when it comprises item 4. Cronbach’s alpha raised from 0.71 to 0.75 for the ‘Physical/ADL’ subscale if item 4 was added, and it remained almost constant for the ‘Echo-coprophenomena’ subscale if item 4 had been deleted (0.83 versus 0.84). “

18. The designation of “Cognitive” seems somewhat misleading, given the nature of the items in that scale, but I guess that might be due to the original English version…

Yes, items from the ‘Cognitive’ subscale are the same in the two versions and we therefore kept the term “Cognitive” for this subscale.

19. In Table 5, the problematic items (9, 10, and possibly 15 too) from the “Obsessive-compulsive” scale should be highlighted somehow, so that the readers can identify them more easily.

Correlations of items with their parent subscale (corrected for overlap) are in bold. A table footnote has been added.

20. In page 20, I think it would be better to specify some of the values instead of only mentioning “small” or “moderate” correlations.

As recommended, values of coefficients were added to the interpretation of the correlation.

21. Are all p-values in Table 7 correct? It would seem more logical if they were 0.01 and 0.001 instead, given that * corresponds to 0.05.

You are right, there was an error with the p-values. They were changed to:

“Correlations significantly different from zero: * p<0.05, ** p<0.01 and *** p<0.001.”

22. Can you please explain if the EuroQol-5D has important flaws that one should be aware? That would help understanding why you suggested a different approach.

A sentence was added to explain why we suggest a different approach than EuroQol-5D. (page 34).

“The EQ-5D, is a very simple tool allowing for a very general assessment of quality of life and possesses only five items, with one item per dimension. It is mostly used in medico-economic studies [49].”

Reference 49: Rabin R, de Charro F. EQ-5D: a measure of health status from the EuroQol Group. Ann Med. 2001;33:337‑43.

23. I do not quite agree with the following sentence: “These observations are evidence that psychological considerations are those that have the greatest influence on the HRQoL of GTS subjects”. Psychological symptoms might themselves be a direct consequence of the severity of Tourette syndrome and/or comorbidities symptoms…

We agree with your remark and changed the sentence (page 36):

“These observations are evidence that psychological considerations (i.e. psychiatric/behavioral comorbidities or direct psychological consequences of the severity of GTS) are those that have the greatest influence on the HRQoL of GTS subjects.”

Typos:

1. There is an extra “)” following “[15]” in the “Translation and cultural adaptation of the GTS-QOL” subsection.

2. There is a blank space missing before “[28]”. 

Corrections have been done.

---

## [Editor Report · Decision Letter 1]

1 Dec 2020

Cross-cultural adaptation and psychometric evaluation of the French version of the Gilles de la Tourette Syndrome Quality of Life Scale (GTS-QOL)

PONE-D-20-21404R1

Dear Dr. Jalenques,

We’re pleased to inform you that your manuscript has been judged scientifically suitable for publication and will be formally accepted for publication once it meets all outstanding technical requirements.

Kind regards,

Sandra Carvalho

Academic Editor

PLOS ONE

---

## [Editor Report · Acceptance letter]

11 Dec 2020

PONE-D-20-21404R1 

Cross-cultural adaptation and psychometric evaluation of the French version of the Gilles de la Tourette Syndrome Quality of Life Scale (GTS-QOL) 

Dear Dr. Jalenques:

I'm pleased to inform you that your manuscript has been deemed suitable for publication in PLOS ONE. Congratulations! Your manuscript is now with our production department. 

Kind regards, 

on behalf of

Dr. Sandra Carvalho 

Academic Editor

PLOS ONE